# A Facile Single-Phase-Fluid-Driven Bubble Microfluidic Generator for Potential Detection of Viruses Suspended in Air

**DOI:** 10.3390/bios12050294

**Published:** 2022-05-03

**Authors:** Jia Man, Luming Man, Chenchen Zhou, Jianyong Li, Shuaishuai Liang, Song Zhang, Jianfeng Li

**Affiliations:** 1Key Laboratory of High Efficiency and Clean Mechanical Manufacture of MOE, School of Mechanical Engineering, Shandong University, Jinan 250061, China; mlm@mail.sdu.edu.cn (L.M.); zhouchenchen161@163.com (C.Z.); ljy@sdu.edu.cn (J.L.); zhangsong@sdu.edu.cn (S.Z.); ljf@sdu.edu.cn (J.L.); 2Key National Demonstration Center for Experimental Mechanical Engineering Education, Shandong University, Jinan 250061, China; 3State Key Laboratory of Tribology, Tsinghua University, Beijing 100084, China; 4School of Mechanical Engineering, University of Science and Technology Beijing, Beijing 100084, China; liangss@ustb.edu.cn

**Keywords:** microfluidic device, single-phase fluid, bubbles, air detection

## Abstract

Microfluidics devices have widely been employed to prepare monodispersed microbubbles/droplets, which have promising applications in biomedical engineering, biosensor detection, drug delivery, etc. However, the current reported microfluidic devices need to control at least two-phase fluids to make microbubbles/droplets. Additionally, it seems to be difficult to make monodispersed microbubbles from the ambient air using currently reported microfluidic structures. Here, we present a facile approach to making monodispersed microbubbles directly from the ambient air by driving single-phase fluid. The reported single-phase-fluid microfluidic (SPFM) device has a typical co-flow structure, while the adjacent space between the injection tube and the collection tube is open to the air. The flow condition inside the SPFM device was systematically studied. By adjusting the flow rate of the single-phase fluid, bubbles were generated, the sizes of which could be tuned precisely. This facile bubble generator may have significant potential as a detection sensor in detecting viruses in spread droplets or haze particles in ambient air.

## 1. Introduction

When the size scale of objects reaches microns or even sub-microns, some unique advantages, such as high surface–volume ratio, superior convenience of delivery, and comparatively high production throughput, make microcomponents suitable for the applications in biomedical engineering [1,2,3], materials synthesis [4,5,6], biochemical detection [7], etc. For example, the droplet digital PCR (ddPCR) technic can discretize reagents into droplets with picolitre or nanolitre volumes for analysis of single cells, organisms, or molecules. [8] Additionally, recently, ddPCR has been reported as a powerful tool for the rapid diagnosis of COVID-19 [9,10]. In addition, microdroplets can also act as the ideal microreactors for chemical reactions, from which varieties of microparticles can be prepared according to the demands for different applications. ZrO_2_ ceramic microspheres prepared from the droplet-based microfluidic method can be the template for UO_2_ microspheres used as fuel for the nuclear power station [11,12]. Al_2_O_3_ ceramic microspheres with controlled structures can be ideal powder for coatings preparation [13]. Calcium alginate microspheres with narrow size distribution and controlled structures can be drug carriers, having controlled drug release profiles in the treatment of acute kidney injury [14,15]. Microbubbles have various applications in drug delivery systems [16] and as ultrasonic contrast agents [17]. Additionally, it was found that focal drug delivery to a vessel wall facilitated by intravascular ultrasound and microbubbles is a promising potential therapy for atherosclerosis [18]. Huang et al. reported ultrasound-responsive microbubbles for the combination tumor treatment [19]. They prepared gas in hydrogel microbubbles with gemcitabine (GEM) encapsulated in poly(N-isopropyl acrylamide) (PNIPAM) shells and hydrogen sulfide cores. It was demonstrated that the release of multiple drug-loaded microcarriers could be controlled by triggering with ultrasound, which has good therapeutic efficacy in tumor treatment.

Conventionally, microbubbles/droplets are produced by using inhomogeneous extensional and shear flows to rupture mother droplets, forming smaller ones. Certainly, it is hard to precisely control the rupture and produce emulsions of monodispersity since the ambient flows are turbulent [20]. Due to the accurate control of the fluids inside the microchannels, the microfluidic method can be utilized to prepare microdroplets/bubbles with controlled structures and superior size monodispersity [21,22,23]. Generally, the structures of microfluidic devices are extremely diverse, including T-junction, flow focusing, co-flow, membrane, and step emulsification [24,25]. However, the mechanism of droplet/bubble formation in the microfluidic device with the above structures is that the liquid–gas threads of the dispersed phase are ruptured under the action of shear forces by the immiscible and continuous phase. For example, Wan et al. reported a microfluidic approach with two different geometries to generate gas-in-water-in-oil (three-phase) emulsions in a PDMS device by changing the liquid flow rates and gas pressure [26]. This gas-in-water-in-oil emulsion could be the template for porous microparticles. Chen et al. used a coaxially capillary microfluidic device to fabricate gas-in-oil-in-water triple emulsions, in which the gas core encapsulated inside can both act as a contrast agent and induce controlled release upon application of ultrasound [27]. Huang et al. used a typical co-flow structured capillary microfluidic device to fabricate H_2_S gas-in-Alginate/NIPAM/PVA emulsions and solidified them in a CaCl_2_ solution to form gas-in-polymer microparticles, which could be used to treat pancreatic tumors by ultrasound triggering [19].

That is to say, in the microfluidic strategy of making microdroplets/bubbles, at least two phases of fluid should be controlled either by syringe pumps [28,29,30] or the gravity of the fluid [31].

In this study, we introduced a novel microfluidic device, in which microbubbles/droplets can be formed only by a driving single-phase fluid. The single-phase-fluid microfluidic device (SPFM) has a similar structure to the typical co-flow microfluidic device. The injection tube of the SPFM device is connected to a syringe driven by a pump. However, the collection tube of the SPFM device is open to the air, which is different from the traditional co-flow structure, as shown in Figure 1a. When the inner-phase fluid is driven into the injection tube of the SPFM device, gas bubbles are formed downstream in the collection tube under certain conditions, as shown in Figure 1b,c. We surprisingly found that the gas bubbles generated in the SPFM device have a narrow size distribution, as shown in Figure 1d. Then, we systematically studied the effects of flow rate and geometric parameters of SPFM on the production of microdroplets/bubbles. Additionally, the bubble formation phenomenon was theoretically investigated. Inspired by the application of ddPCR in the diagnosis of COVID-19, the SPFM device can transform the air from an ambient atmosphere into gas bubbles with homogeneous volume, which may have great potential in the detection of viruses or pollutants suspended in the ambient air.

## 2. Experimental Methods

### 2.1. Materials

DI water (Sigma-Aldrich, SigmaAldrich, St. Louis, MO, USA) was used as the inner-phase fluid in the experiment of making air bubbles in the SPFM device. Briefly, 1wt% SDS (Meilun BioTech Co., Ltd., Dalian, China) was added to the DI water to prevent the coalescence of the bubbles. Silicone oil (Aladdin Reagent Shanghai Co., Ltd., Shanghai, China; 20 mPa.s) with 1% *v*/*v* Span 80 (Sinopharm Chemical Reagent, Beijing, China) was used as the oil phase to make monodispersed aqueous and oil microdroplets.

### 2.2. Setup of SPFM Device

The SPFM device has a typical co-flow structure, which was composed of two cylindrical glass capillary tubes called an injection tube and a collection tube, respectively. The injection glass tube with an inner diameter of 300 µm was coaxially positioned inside the collection tube with an inner diameter of 900 µm. To maintain the vertically coaxial position of the two capillary tubes, we placed four papers (about 0.4 mm) under the injection tube. Additionally, to maintain the horizontally coaxial position of the two capillary tubes, we adjusted their position under a microscope manually. Two glass tubes were fixed on the glass sheet with epoxy glue, and the entrance of each glass tube was connected with a dispensing needle, as shown in Figure 1a. However, unlike the traditional co-flow microfluidic device, the orifice between the injection tube and collection tube was open to the air. Moreover, the surface wettability of the collection tube was tuned according to the bubbles/droplets made in the SPFM device. When making monodispersed bubbles and oil droplets, the collection tubes were hydrophilic, and therefore, they did not need to be further treated. When making aqueous droplets, the inner surface of the collection tube had to be wet during the oil phase, and therefore, it was treated to be hydrophobic by immersing in an octadecyl trichlorosilane solution (OTS, Sigma-Aldrich, Shanghai, China) for 10 s.

### 2.3. Preparation of Microbubbles/Droplets in the SPFM Device

Making microbubbles in the SPFM device is relatively simple, as we only need to set the flow rate of the inner-phase fluid under the critical value. A syringe pump (Longer Pump LSP01-3A, Baoding, China) was used to inject the fluid into the SPFM device. When making aqueous/oil droplets using the SPFM method, a reservoir needs to be connected at the orifice between the injection tube and the collection tube. The entire process of bubble formation in the experiment was carried out in a 25 °C environment.

### 2.4. Detection of the Fluorescent Microparticles in Ambient Air

Nile red nanoparticles (100 nm, EKEAR BioTech Co. LTD, Shanghai, China) were suspended in paraffin oil and then atomized by an aerosol generator. An acrylic box was used to simulate the space to be detected. The SPFM device was placed inside the sealed acrylic box, into which the atomized spread droplets with fluorescent nanoparticles were injected. Then, the flow rate of the inner-phase fluid was set as 3 mL/min. Bubbles with fluorescent nanoparticles inside were generated in the collection tube of the SPFM device and collected in a Petri dish. Then, images of the bubbles were captured in a fluorescent microscope (Leica DM4B, Leica Camera, Wetzlar, Germany), and the fluorescent intensity of the bubbles was measured using ImageJ software (v1.50i, National Institutes of Health, Bethesda, MA, USA).

## 3. Results and Discussion

### 3.1. Microbubble Formation and Size Control

In the SPFM device, only the inner-phase fluid was pumped into the injection tube with a controlled flow rate. The continuous phase is the gas in ambient air, and the inlet of the collection tube is directly open to the air, as shown in Figure 1a,b. We injected DI water into the injection tube of a vertically positioned SPFM device, at a flow rate of 5 mL/min, and observed the flowing condition in the collection tube with a horizontal microscope with a high-speed camera. When the water flowed into the collection tube from the orifice of the injection tube, the flowing speed of the water stream decreased immediately due to the difference between the diameters of the injection and collection tubes. The collection tube is hydrophilic; thus, the water stream adheres to the inner surface of the microchannel, acting as the real continuous phase in the collection tube. The gas from the ambient air is sucked into the collection tube from the adjacent space between the injection tube and the collection tube when the flow rate of water is in a certain range, forming a long thread under the viscous force of water and eventually splitting into bubbles due to the Rayleigh Plateau instability [32], as shown in Figure 1c. The bubble formation process is shown in the Appendix A. The throughput of the bubble generation could be around 500 per hour.

We systematically studied the effects of the flow rate of the water on the sizes of the formed microbubbles. Figure 2a shows that the diameter of microbubbles formed in the SPFM device can be adjusted by changing the flow rate of the inner-phase fluid in a certain range. The improper choice of the flow rate would not lead to the bubble formation phenomenon, which is discussed in the following section. It can be found in Figure 2a that the diameter of microbubbles decreased with the increase in the flow rate. Figure 2b,c show the images of microbubbles with size distributions formed at 1 mL/min and 5 mL/min, respectively, from which we can conclude that the size distribution of the microbubbles formed in the SPFM device was narrow. This indicated that the SPFM method can be used to make monodispersed microdroplets/bubbles.

### 3.2. Mechanism of Microbubble Formation

As described above, the bubble formation phenomenon can only occur in a certain situation. For the SPFM device with a fixed size, when the flow rate of the water inside the injection tube is set at a relatively high value, water fills the microchannels and leaks outside from both the upper and lower ends of the collection tube. Therefore, we systematically decreased the flow rate of the inner-phase fluid, and bubbles appeared when the flow rate was lower than a certain flow rate value, which was recorded as the critical flow rate of the bubble formation phenomenon.

Then, we studied the effects of geometric parameters of the SPFM device and the flow rate on the bubble formation phenomenon. The length of the injection tube inserted in the collection tube, *l*, the length between the orifice of the injection tube and the outlet of the collection tube, *L*, the diameter of the collection tube, *D*, and the tilt angle of the SPFM device, *θ*, were chosen as the geometric parameters to be studied in this research, as shown in Figure 3a. First, we prepared the SPFM devices with different *L* values while keeping other parameters constant and initially flowed DI water into the injection tube at a relatively high flow rate (>7 mL/min). Then, we gradually decreased the injection flow rate and observed the flowing condition inside the collection tube, according to which the bubble formation phase diagram is presented in Figure 3b. The bubble formed in the collection tube when the flow rate was reduced to a certain flow rate *Q*, which was recorded as the critical flow rate for the bubble formation phenomenon. Each experiment on the critical flow rate *Q* was repeated at least three times, and it was found that, when the flow rate was higher than the critical *Q*, DI water filled the collection tube, and no bubbles were generated. When the flow rate was under the critical value, bubbles formed in the SPFM device. Similarly, we systematically investigated the effects of different parameter combinations, *Q* and *l*, *Q* and *D*, *Q* and *θ*, on the bubble formation phenomenon, and summarized the experimental results in phase diagrams, which were divided into two different regions, as shown in Figure 3c–e. When the flow rate was larger than the critical *Q*, the flow rate point was located in the “No Bubble” region, as shown in the diagram, indicating that no bubbles were formed at this flow rate, while when the flow rate was adjusted to a smaller value than the critical value, bubbles were formed inside the SPFM device. This phenomenon was attributed to the siphon effects inside the collection tube. When DI water flowed into the collection tube from the orifice of the injection tube, the flowing velocity of the DI water stream decreased rapidly due to the different diameters of the injection and collection tube. Then, the DI water flowed downstream along the collection tube, while the gravity of the water also affected the flowing condition of the DI water inside the collection tube. Assuming the starting position of the DI water flowing inside the collection tube is position 1, while the exit position of the collection tube is position 2, as shown in Figure 3a, we could obtain the following relation according to the Bernoulli equation:(1)$$L\cos\theta +\frac{P_{1}}{\rho g}+\frac{v_{1}{}^{2}}{2g}=\frac{P_{2}}{\rho g}+\frac{v_{2}{}^{2}}{2g}+\lambda \frac{L}{D}\frac{{\bar{v}}^{2}}{2g}+A$$
where *L* is the length of the DI water stream inside the collection tube, *D* is the diameter of the collection tube, *P*_1_ and *v*_1_ were the pressure and velocity at position 1, *P*_2_ and *v*_2_ were the pressure and velocity at position 2, *θ* is the angle between the collection tube and the directional gravity, and g is the acceleration of gravity. *A* is the correction factor in this experiment, which represents the energy loss caused by the random factors. *λ* is the friction coefficient considering the pressure drop due to the interactions between the inner surface boundary and the fluid [33].
*λ* = 64*K*/*Re*(2)
where *K* is the friction factor, which represents the energy loss caused by the friction between DI water and the inner surface of the collection tube. *Reis* the Reynolds number, Re=ρvD/u. *u* = 10^−3^ Pas.

When the DI water flowed inside the collection tube, the gravity of DI water caused an increase in the flow rate of the water stream inside the collection tube. When the injecting flow rate was high enough, the water stream inside the injection tube could replenish the collection tube, although the flow rate of DI water in the collection tube had already been accelerated by gravity. When we gradually decreased the injecting flow rate to a critical value at which the injecting DI water could not maintain the continuous flow of DI water inside the collection tube, bubbles were formed. We assumed that, at the critical condition of the bubble formation phenomenon, the injecting flow rate is equal to that of the gravity-accelerated-flow rate. The pressure at positions 1 and 2, *P*_1_ and *P*_2_ is the atmospheric pressure [34]. Additionally, in this experiment, *v*_1_ << *v*_2_; thus, we assumed *v*_1_ ≈ 0. In the case of considering the effects of length *L* on the bubble formation, *L* varied from 0 to 100 mm, and *D* was 0.9 mm; therefore, we can obtain the fitting line from Equation (1) using the nonlinear implicit function fitting in Origin software, which agrees well with the experimental results, as shown in Figure 3b. In this situation, the friction factor *K*_L_ ≈ 0.98 ± 0.027, the correction factor *A**_L_* ≈ 0.03 ± 0.00065, and the coefficient of determination, *R_L_*^2^ = 0.9999, which indicates the equation fits well with the experimental data. Similarly, in the case of considering the effects of *D* and *θ*, we can also obtain the fitting lines, which agree well with the experimental critical *Q*, as shown in Figure 3c,d. Additionally, the friction factor *K_D_* ≈ 1.61 ± 0.267, *A**_D_* ≈ 0.015 ± 0.012, R*_D_*^2^ = 0.9997, *K**_θ_* ≈ 0.70 ± 0.041, *A**_θ_* ≈ 0.045 ± 0.000041, *R_θ_*^2^ = 0.9999, respectively. Using Equation (1), it was found that the length of the injection tube inside the SPFM device does not influence the bubble formation. Thus, the critical *Q* for bubble formation was maintained at a constant value while changing the injection tube length and keeping other parameters constant, as shown in Figure 3d.

### 3.3. Droplet Generation in the SPFM Device

The SPFM device can also be used to produce monodispersed droplets besides making bubbles. The structure of the droplet-making SPFM device is similar to that of the bubble-making SPFM device, and only a reservoir was added to the inlet of the collection tube, as shown in Figure 4d,h. The SPFM device can be used to produce both aqueous droplets and oil droplets, only by changing the surface wettability treatment of the collection tube. When making droplets, only the flow rate of the inner-phase fluid was controlled. The sizes of the droplets can be tuned by the flow rate of the inner-phase fluid. It can be found that the diameters of the droplets decreased with the increase in the flow rate, as shown in Figure 4a,e. The microdroplets prepared by the SPFM device had a narrow size distribution, as shown in Figure 4b,c,f,g. Thus, the SPFM device can be used to prepare gas, aqueous, and oil microdroplets with a narrow size distribution, only by controlling the flow rate of the single-phase fluid.

### 3.4. Potential Application in Virus Detection

From the above description, the SPFM device divides the gas/solution connected with the adjacent space between the injection tube and collection tube into monodispersed units. This phenomenon may have great potential in the detection of haze or viruses suspended in the air. To verify the detection capacity of viruses suspended in the air, a simulated experiment setup is illustrated in Figure 5a. The SPFM device was placed in a sealed acrylic box, into which an aerosol generator injected atomized droplets. To simulate the size of the COVID-19 virus [35], 100 nm sized fluorescent Nile Red nanoparticles were used to simulate the virus within the exhaled droplets since it was difficult to receive the required qualifications to perform virus-spread experiments. As introduced above, when the water was flown into the injection tube of the SPFM device, bubbles with simulated spread droplets were formed in the collection tube. Then, the collected bubbles were observed using fluorescent microscopy, as shown in Figure 5b. It can be found that bubbles with a higher Nile Red concentration (Figure 5(b2)) were brighter than those with a lower concentration (Figure 5(b1)) in the fluorescent field. Then, we systematically investigated the relationship between the concentration of Nile red suspended in the spreading droplet and the fluorescent intensity. Additionally, it should be noted that the bright spot in each of the bubbles in Figure 5b (at about 10 O’clock) is due to the reflection or refraction of light. From Figure 5c, we can find that the fluorescent intensity tended to increase with the increase in Nile Red concentration, which indicates that the SPFM device may have the potential to be used to detect viruses or haze particles in the air.

## 4. Conclusions

In summary, we introduced a single-phase-fluid-driven microfluidic (SPFM) device, which could generate monodispersed bubbles/droplets by controlling only the flow rate of the injection phase fluid. The effects of geometrical parameters of the SPFM device and the flow rate of inner phase fluid on the flowing condition inside the device were systematically studied. It was found that only under a critical flow rate value can the bubble formation phenomenon occur, which can be explained by the Bernoulli equation. Furthermore, the bubble formation phenomenon that occurred in the SPFM device can be used to detect the concentration of suspended fluorescent particles in spread droplets in the air. This may have significant potential in making monodispersed bubbles/droplets from an ambient environment or in the detection of haze or viruses in the air.

## Figures and Tables

**Figure 1 biosensors-12-00294-f001:**
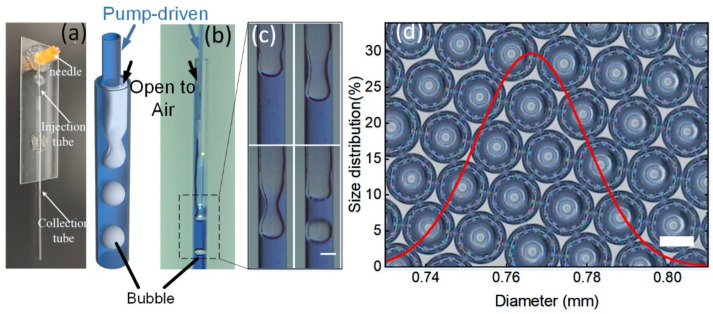
(**a**) Images of the SPFM device; (**b**) schematics illustration and images captured in microscopes of the formation of the microbubbles in the SPFM device, which has a typical co-flow structure. However, different from the currently reported co-flow microfluidic method, water was pump-driven into the inner tube, while the collection tube was open to the air, and (**c**) microbubbles were formed downstream in the collection tube; the scale bar is 500 μm; (**d**) microbubbles prepared in the SPFM device have good monodispersity; the scale bars are 500 μm.

**Figure 2 biosensors-12-00294-f002:**
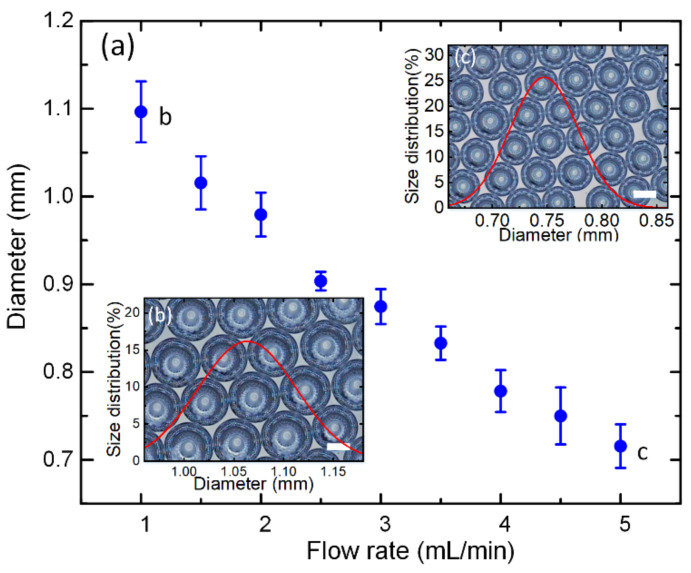
(**a**) The diameter of microbubbles generated in the SPFM device (collection tube, I.D. = 900 μm) can be tuned by adjusting the flow rate of the water phase; (**b**) large bubbles with good monodispersity were generated at a relatively small flow rate, 1 mL/min; scale bar is 500 μm; (**c**) small microbubbles generated in a relatively large flow rate, 5 mL/min, have a narrow size distribution. The scale bar is 500 μm.

**Figure 3 biosensors-12-00294-f003:**
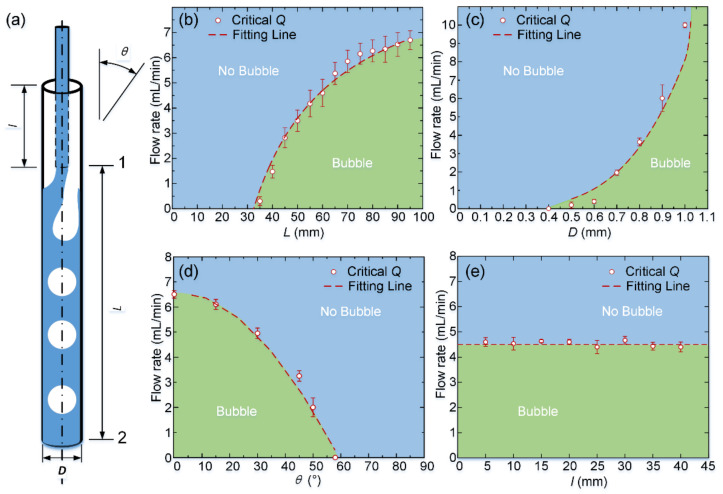
(**a**) Schematic diagram of the SPFM device. The effects of geometry parameters of the SPFM device, *θ*, *L*, *D*, and *l* on the bubble formation were systematically evaluated; (**b**–**d**) the phase diagram of bubble formation phenomena. The blue zone corresponds to the regime with no bubble formation in the device, while the green zone means microbubbles can be prepared under these parameter combinations. The dashed line is the theoretically fitted curve, and the red circles represent the critical flow rate for bubble formation while changing the geometry parameters: (**b**) *L*, (**c**) *D*, (**d**) *θ*, and (**e**) *l*.

**Figure 4 biosensors-12-00294-f004:**
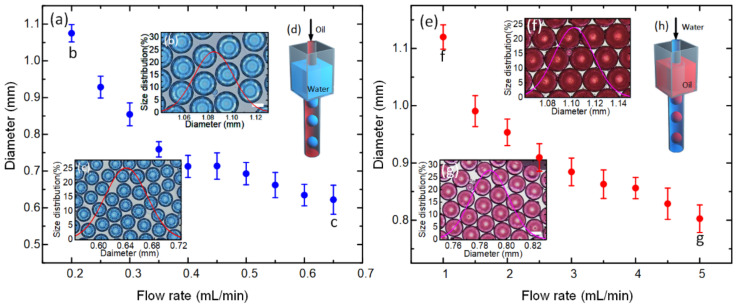
Formation of aqueous and oil microdroplets in the SPFM device: (**a**) relationship between the diameter of aqueous droplets and the flow rate of aqueous fluid; (**b**) images of monodispersed microdroplets prepared at 0.2 mL/min; the scale bar is 500 μm; (**c**) images of monodispersed microdroplets prepared at 0.65 mL/min; the scale bar is 500 μm; (**d**) schematic diagram of the formation of aqueous microdroplets in the SPFM device; (**e**) relationship between the diameter of oil droplets and the flow rate of oil fluid; (**f**) images of monodispersed microdroplets prepared at 1 mL/min; the scale bar is 500 μm; (**g**) images of monodispersed microdroplets prepared at 5 mL/min; the scale bar is 500 μm; (**h**) schematic diagram of the formation of oil microdroplets in the SPFM device.

**Figure 5 biosensors-12-00294-f005:**
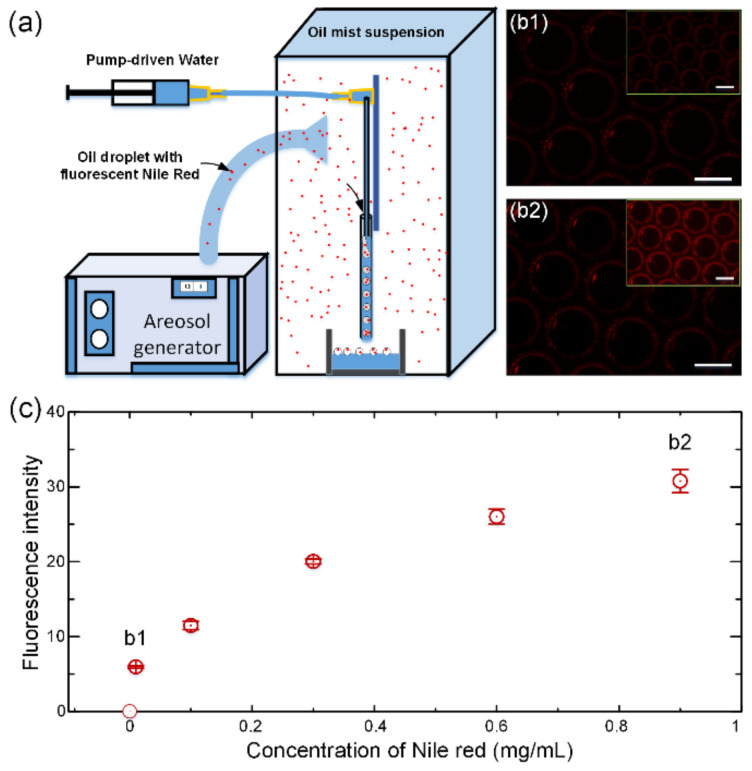
(**a**) Schematics of air detection based on the SPFM method. The fluorescence dye, Nile Red, was suspended in paraffin oil. The suspension was then atomized into a sealed chamber by an aerosol generator. The SPFM device inside the chamber was used to make microbubbles from the air with fluorescence dyed oil mist; (**b1**) images of microbubbles prepared from oil mist with 0.01 mg/mL Nile Red; (**b2**) images of microbubbles prepared from oil mist with 0.01 mg/mL Nile Red. The inlets in (**b1**,**b2**) are the same images with 40% brightness and contrast enhancement for better observation in the article. The scale bars are 500 μm; (**c**) the relationship between the concentration of Nile Red suspended in oil mist and the fluorescence intensity of the microbubbles captured using fluorescent microscopy.

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
