# Peer review of "A Facile Single-Phase-Fluid-Driven Bubble Microfluidic Generator for Potential Detection of Viruses Suspended in Air"

_biosensors, 2022, doi:10.3390/bios12050294_

Round 1

Reviewer 1 Report

Jia et al reported a single-phase fluid microfluidic device in this paper. Variables such as flow rate and geometry parameters were experimentally studied to provide the design rules to generate microbubbles and how to tune their diameters.  The experimental data was compared with the theoretical equation [1], which had good agreement. I believe this paper reported an attractive approach to generate monodispersed bubbles/droplets by controlling only the flow rate of the injection phase fluid. However, some information is missing.

Could you elaborate more details about how did you perform the fitting using the equation [1]. For instance, how did you measure the P1, P2, v2. Is vbar same as the input flow rate. If not could you specify it. Also, how did you calculate the friction factor/correction factors, such as by solving ODEs? 

What is the throughput using this device. for example, how many droplets can be generated per hour. 

How stable the microdroplets are. Will surfactant help to stabilize the microdroplets? How easy you can manipulate microdroplets such as mixing etc.

The authors mention the Reynolds number contributes to the friction coefficient. How much Reynolds number or any other dimensionless numbers contribute to determine the critical Q.  

Reviewer 2 Report

This paper described an approach to make monodispersed microbubbles directly from the ambient air by driving single-phase fluid. The process of making the device should be described in more detail. Some questions need to be answered.

  1. More background and references should be added in the introduction for the coaxially positioned device to generate bubbles or droplets.
  2. How to make sure the two capillary tubes coaxially positioned?
  3. “fixed on the glass sheet with epoxy glue”,what is glass sheet, Figure 1 a is not clear enough. Specifically describe how the two glass tubes were bonded and where was the contact position. How to ensure that the two glass tubes are still coaxial in the bonding process.
  4. Where is dispensing needle, is it in Figure1?
  5. 100nm nanoparticles were used in the experiment, why this size was used. Could you compare the other size nanoparticles?
  6. Effect of the geometry parameters was studied in Figure 3. How about the diameter of the injection tube, will it affect the bubbles or droplets?

Reviewer 3 Report

The manuscript describes the generation of gas-in-water, oil-in-water, and water-in-oil droplet generation using a simple device in which only one of the phases needs to be pumped. The authors present a very thorough characterisation of the generation of droplets of different sizes and propose a theoretical model that describes the performance nicely in terms of the experimental parameters which will be very useful in designing devices for specific performance needs. The authors also show the use of the approach to trap 100nm beads in an aerosol, inside bubbles and propose that this would be useful for virus detection.

The paper is very relevant, on the whole well written and I would be happy to see it published in Biosensors after minor revisions.

The structure of the manuscript is very good, and the descriptions easy to follow, however the language could be improved by the attentions of a native English speaker.

There is no mention of temperature in the manuscript but the viscosity of water is a function of temperature. The fits to the model look good with the viscosity set to 1 mPa s but it would be good to know what the temperature was during experiments.

The application of the device to the capture of aerosolised particles is interesting. It would be good if the authors could give some information about the size of the aerosol particles. Also, the fluorescence signal seems to be coming almost entirely from the interface between the air bubble and the water. This is not so strange since the diffusion length of the aerosol is probably sufficient to ensure that most particles hit the walls within a short time (again, would be good to know the size) and would accumulate there. This is not such a problem for oil, which is immiscible in water, and will stay at the interface, but for hydrophilic viruses it they might be lost to the carrier fluid. There also appears to be a bright spot in each of the bubbles in figure 5 (at about 10 O’clock). Is this due to reflection or refraction of the light used to exit the fluorescence? A comment would be good.    

Round 2

Reviewer 1 Report

Authors have addressed all of my concerns with the original manuscript. With the actual device image, suggested by another reviewer, the manuscript would be much clear. 

Author Response

Thanks for your suggestions. We have added the actual device image in Fig. 1 in the revised manuscript. 

Reviewer 2 Report

Accept in present form

Author Response

Thank your very much for your help to improve our manuscript.